# Current and Future Pathotyping Platforms for *Plasmodiophora brassicae* in Canada

**DOI:** 10.3390/plants10071446

**Published:** 2021-07-15

**Authors:** Heather H. Tso, Leonardo Galindo-González, Stephen E. Strelkov

**Affiliations:** Department of Agricultural, Food and Nutritional Science, University of Alberta, Edmonton, AB T6G 2P5, Canada; htso@ualberta.ca (H.H.T.); galindo@ualberta.ca (L.G.-G.)

**Keywords:** *Brassica napus*, clubroot, *Plasmodiophora brassicae*, pathotyping, diagnostics, PCR, sequencing

## Abstract

Clubroot, caused by *Plasmodiophora brassicae*, is one of the most detrimental threats to crucifers worldwide and has emerged as an important disease of canola (*Brassica napus*) in Canada. At present, pathotypes are distinguished phenotypically by their virulence patterns on host differential sets, including the systems of Williams, Somé et al., the European Clubroot Differential set, and most recently the Canadian Clubroot Differential set and the Sinitic Clubroot Differential set. Although these are frequently used because of their simplicity of application, they are time-consuming, labor-intensive, and can lack sensitivity. Early, preventative pathotype detection is imperative to maximize productivity and promote sustainable crop production. The decreased turnaround time and increased sensitivity and specificity of genotypic pathotyping will be valuable for the development of integrated clubroot management plans, and interest in molecular techniques to complement phenotypic methods is increasing. This review provides a synopsis of current and future molecular pathotyping platforms for *P. brassicae* and aims to provide information on techniques that may be most suitable for the development of rapid, reliable, and cost-effective pathotyping assays.

## 1. Introduction

### 1.1. Clubroot Disease

Clubroot, caused by the obligate parasite *Plasmodiophora brassicae* Woronin, is one of the most serious soilborne diseases of crucifers worldwide. Crop losses due to clubroot are estimated at 10–15% globally [1]. It has become a major threat to the $29.9 billion Canadian canola (*Brassica napus* L.) industry [2], as severely infected canola crops can sustain yield losses of 30% to 100% [3]. The main symptom associated with clubroot is the formation of galls on the host roots, restricting the absorption of water and nutrients needed to support aboveground growth [1]. Infection by *P. brassicae* reduces canola yield and oil quality as plants become stunted and undergo accelerated flowering and premature senescence [4].

Clubroot was first identified on canola in western Canada in 2003 in 12 fields near Edmonton, Alberta [5]. Annual clubroot surveys began in 2004 to monitor the spread of *P. brassicae* and the severity of the disease [6]. As the number of clubroot-infested fields began to increase, *P. brassicae* was declared a pest under the Agricultural Pests Act of Alberta in April of 2007 [7]; this designation enabled the enforcement of control measures throughout the province aimed at reducing the dissemination and impact of the disease. Nonetheless, *P. brassicae* has continued to spread and is considered to be one of the most significant problems facing canola growers. Clubroot is now endemic to much of central Alberta [1], with more than 3000 confirmed field infestations [8] and the disease having been identified in 44 of the 66 counties or municipal districts where canola is grown [9]. Epidemiological models had predicted that clubroot could spread beyond Alberta throughout the Canadian Prairies, and since then the disease has been confirmed on canola in Saskatchewan and Manitoba [3]. Clubroot now also occurs in canola crops in Ontario, Canada [10], and in North Dakota, USA [11].

### 1.2. Emergence of ‘New’ Virulent Pathotypes

The first clubroot-resistant (CR) canola cultivar became available to farmers in 2009, followed quickly by other cultivars from various seed companies. With 40 registered CR cultivars currently on the market, genetic resistance has become the most important tool for the management of clubroot in Canada [12]. In contrast, other practices recommended as part of an integrated clubroot management plan, including longer rotations out of canola and the sanitization of field machinery [13], have not been adopted as widely [14]. Genetic resistance to clubroot relies primarily on single major genes, effective against specific pathotypes or races of *P. brassicae* [15]. Hence, these cultivars can only provide strong qualitative resistance to a select few pathotypes, while having no effect on other pathotypes [16]. This makes them vulnerable to pathotype shifts, which can occur when *P. brassicae* populations are exposed repeatedly to the same resistance source [17]. Indeed, only four years following the commercial release of the first CR cultivar, ‘new’ *P. brassicae* pathotypes able to overcome resistance were detected in Alberta [18]. The 2013 Alberta-wide clubroot survey found greater than expected disease severity in six fields deployed with CR cultivars [19]. Four *P. brassicae* populations from two of these fields caused significantly increased levels of clubroot in greenhouse trials, and three of these populations were highly virulent across six CR canola cultivars [18]. Sixty-one field populations collected in 2014–2016 could overcome resistance in at least one CR cultivar [20]. In 2017 and 2018, nine novel pathotypes were discovered throughout western Canada [21], and another four novel pathotypes were discovered in the Peace Country Region of Alberta alone [22]. An additional six novel pathotypes were identified during an investigation of *P. brassicae* single-spore isolates collected from field populations that were virulent on CR cultivars [23]. This suggests an increasing diversity in the virulence of *P. brassicae* strains and a greater prevalence of resistance-breaking pathotypes. While the resistance is often said to have “broken down”, the change occurred in the pathotype structure of the *P. brassicae* populations and not in the host cultivars themselves [15]. The repeated cultivation of CR cultivars imposed selective pressure that led to virulence shifts in the pathogen, encouraging the proliferation of novel virulent pathotypes [13].

### 1.3. Need for Rapid Pathotyping

Clubroot is a “disease of cultivation” due to its correlation with intensive cultivation of susceptible crucifers [1]. The evolving nature of *P. brassicae* populations and the development of new canola cultivars illustrate the importance of interdisciplinary efforts to synchronize progress in clubroot management systems. With the emergence of new pathotypes overcoming resistance, the need to distinguish rapidly between pathotypes has become a priority in the development of clubroot diagnostics. It is important for growers, the industry, and researchers to understand the distribution and occurrence of pathotypes in order to make informed crop management decisions. Given the physiologic specialization in *P. brassicae*, great variation can occur in resistance to clubroot among different canola cultivars. Early pathotype identification allows canola farmers to select cultivars that carry the appropriate resistance and provide the best protection against the disease. The deployment of resistance effective against pathotypes found in a specific field or region is essential to reducing disease spread and facilitating preventative management practices. In addition, reliable information on the abundance and diversity of existing pathotypes may encourage breeding programs to target development of CR canola cultivars with the appropriate resistance traits. Since pathotype identification before and during the cultivation of a crop would help to improve clubroot management, a rapid molecular detection method is fundamental. The objective of this review is to assess current and future pathotyping platforms for *P. brassicae*.

## 2. Current Diagnostic Methods

Several diagnostic methods exist for the detection of *P. brassicae* and for the differentiation of *P. brassicae* pathotypes. Although each method has its own limitations, collectively, they all serve a role in the clubroot diagnostic process.

### 2.1. Phenotypic Approaches

Soil bioassays with bait crops have long been used for the general detection of *P. brassicae*. Bait crops are host plants that stimulate resting spore germination and can become infected. One of the first reported methods used cabbage (*B. oleracea*) hosts grown in suspect soil, with the root hairs inspected for the presence of zoosporangia via microscopy after one week [24]. This same method was further used to evaluate resting spore survival in the soil [25]. To ensure that *P. brassicae* detection correlated with gall development, another method grew host plants in the suspect soil, with the roots examined for galling after five weeks [26,27]. While bait crops provide reliable viability assessments of *P. brassicae* inoculum in infested soil, these methods are only feasible when inoculum concentrations are greater than 1000 spores per gram of dry soil, as this is the concentration generally required for gall development under greenhouse conditions [28]. Moreover, soil bioassays may not always be practical, since they are time-consuming and labor-intensive, and require large amounts of greenhouse space. The planting of a single, susceptible bait crop also does not provide information on the race or pathotype classification of the inoculum present.

*P. brassicae* pathotypes are currently distinguished phenotypically based on their virulence patterns in bioassays conducted with host differential sets, where the reactions of the hosts are monitored based on root gall development. These assays aim to capture the occurrence and extent of physiologic specialization in populations of the pathogen [16]. Numerous differential sets have been proposed to study pathogenic variability in *P. brassicae* [20,29,30,31,32]. The differentials of Williams, one of the earliest and among the most commonly used systems, consists of four differential hosts that can distinguish a theoretical maximum of 16 pathotypes [29]. The European Clubroot Differential (ECD) Set was established later and consists of 15 differential hosts, three of which belong to Williams’ system [30]. Somé et al. developed another differential set consisting of three *B. napus* genotypes [31]. The development of the Canadian Clubroot Differential (CCD) set was deemed necessary after the discovery of new virulence phenotypes that overcame resistance in Alberta, Canada [20]. The CCD set consists of 13 host genotypes; these include the differentials of Williams [29] and Somé et al. [31], eight hosts from the ECD set, and several *B. napus* cultivars of interest to Canadian canola breeders [20]. Since the differentials of Williams [29] and Somé et al. [31] are incorporated entirely into the CCD set, pathotype characterization against these earlier sets is possible with the CCD set (Figure 1). Isolates of *P. brassicae* are inoculated onto each host of the CCD set, and disease development is assessed six weeks later [20]. Pathotypes are identified based on the reaction of the inoculated hosts, with distinctive virulence patterns representing individual pathotypes. Each pathotype is assigned a number followed by a letter (for example, ‘pathotype 3A’); the numbers correspond to the Williams’ [29] pathotype designation, while the letters are unique to the CCD set and denote variants of the Williams’ pathotypes [20,23]. Most recently, the Sinitic Clubroot Differential (SCD) set was developed in China using differential hosts with known clubroot-resistance genes to explore the genetic variability of pathotype 4, as defined by the differentials of Williams [32]. The SCD set was essential when isolates identified as pathotype 4 on the system of Williams were found to exhibit varied virulence patterns on various Chinese cabbage (*B. rapa*) and cabbage (*B. oleracea*) cultivars.

Although the CCD set is effective and has greater differentiating capacity than earlier differential systems, several limitations are associated with the use of differential host sets in general. The methodology is labor-intensive, and sufficient biosecure greenhouse space is required due to the need for replication and pathogen containment. Moreover, the evaluation of pathotypes using differential sets is time-consuming, and the generation of virulence patterns can be affected by the greenhouse conditions and biological growth of the root galls and differential hosts. Typically, the differential hosts are rated about 6 weeks following inoculation, and hence only a comparatively small number of isolates can be tested simultaneously, depending on space availability. Therefore, it is difficult to scale up this methodology for rapid testing or for testing large numbers of samples. There is also a risk of human error and variable results among diagnostic laboratories considering that the identification of virulence patterns requires a high degree of technical expertise [20]. Moreover, the existing clubroot differential systems are based on phenotypic classifications, which may not match genomic variation among pathotypes. Since the genetic basis of resistance in some of the host genotypes is not well defined, it may be difficult to infer genetic relationships between isolates or the specific avirulence/virulence genes found in an isolate based on the pathotype classification. This may be addressed through the development of near-isogenic lines carrying defined resistance genes for use as differentials, to more appropriately define true ‘races’ based on the genetics of the host-pathogen interaction. The emergence of new virulence types of *P. brassicae* may also require modification of the differential sets, since ultimately the ability to distinguish pathotypes or races of a pathogen is limited by the effectiveness of the differential hosts.

### 2.2. Microscopy

Histological approaches have long been used to study clubroot. Various staining methods have been employed to visualize *P. brassicae* and host structures. Resting spores can be distinguished from root tissue using a triple staining method [33]; resting spores stain blue, whereas the host tissue stains pink or purple. Another method uses lactophenol cotton blue to stain chitin, a prevalent polymer in the cell wall of *P. brassicae* resting spores [34]. Using methylene blue, resting spores inside the secondary plasmodia are stained a dark blue, while the cell wall of the root tissue stains a contrasting light blue [35]. Primary plasmodia and zoospores can be observed with Harris hematoxylin staining followed by a counterstain with eosin Y [36]. Toluidine blue is used to detect the host resistance response during infection, by staining lignified cell walls blue, since lignification occurs as a defense mechanism [37].

Microscopy has also been used to assess inoculum load and viability as well as germination rates. The viability of resting spores can be tested with Evan’s blue, in which cell membranes of the dead spores stain blue [38,39], and with acridine orange fluorescent dye, where the viable spores fluoresce green [40]. Distinctive fluorochromes for differential staining are also used; non-viable spores fluoresce red whereas viable spores fluoresce blue [41]. Germination levels can be estimated with aceto-orcein, which stains ungerminated resting spores [42].

While histological investigations are fundamental for diagnosing the disease as a whole and advancing clubroot research, preventative measures for clubroot remain a challenge, as microscopy-based techniques are not sensitive enough for early pathogen detection. More importantly, they are of no value for pathotyping since cell morphology does not differ between pathotypes.

### 2.3. Molecular Approaches

The development of methods for the molecular detection of *P. brassicae* has been an ongoing process over the last three decades (Table 1). A PCR assay was developed for the general detection of *P. brassicae* in soil [43]. The primers were designed based on an isopentenyltransferase-like gene specific to *P. brassicae*. To increase sensitivity, another PCR assay was developed based on an internal transcribed spacer (ITS) region of ribosomal DNA [44]. Since there are more copies of this ITS region in the *P. brassicae* genome, detection can occur at lower resting spore concentrations. After initial tests of these primers in artificially infested soils, the primers were later incorporated into an assay for the detection of *P. brassicae* in naturally infested field soils [45]. The main drawback of these assays [43,44,45] was their nested PCR design, which involves two amplifications. This requires more time and materials, and there is an increased risk of contamination during sample manipulation in comparison with one-step PCRs. After the start of the clubroot outbreak in western Canada in 2003, a non-nested, one-step PCR assay was developed for the molecular detection of *P. brassicae* [46]. The PCR primers were based on a conserved 18S ribosomal RNA gene that could detect the pathogen in symptomless root tissues as early as three days after inoculation and with concentrations as low as 1 × 10^3^ resting spores per gram of soil. This assay has provided the groundwork for molecular testing of clubroot in Canada, and has since been commercially available from diagnostic laboratories throughout the country [28]. Although these reported PCR assays are effective for clubroot detection, they were initially non-quantitative and therefore could not provide information on levels of soil inoculum or host colonization.

The development of quantitative PCR (qPCR) assays for clubroot detection has been reported over the last decade. In qPCR assays, the initial amount of pathogen DNA is directly correlated with an early or late exponential curve of amplification. Initially, two dye-based assays were reported, one for the quantification of resting spores in plant samples [47], and another for the quantification of resting spores in seeds harvested from infested fields [48]. Both assays were successful in rapid quantification; however, dye-based qPCR is less specific in comparison with probe-based qPCR. Only one target can be investigated at a time with dye-based technologies, as the dye will bind to any DNA fragment amplified in the reaction; therefore, these experiments rely on careful primer design and amplicon selection. In probe-based qPCR, a fluorogenic probe anneals to a specific sequence within the PCR amplicon during the reaction. Two probe-based assays were developed to quantify resting spores in naturally infested soil samples [49,50], and another was developed for the quantification of *P. brassicae* in root tissue [51]. Since probe-based technologies have greater specificity and multiple targets can be detected simultaneously in each sample, they may be more suitable for clubroot quantitative diagnostics. Up to this point, qPCR assays cannot discriminate viable from non-viable resting spores. To address this issue, propidium monoazide (PMA) was incorporated into a probe-based qPCR assay to prevent the amplification of non-viable resting spores [52]. PMA is a photoreactive DNA-binding dye that penetrates the cells of dead membranes and is commonly used for viable microorganism distinction. An assay was designed using droplet digital PCR (ddPCR) to quantify resting spores in soil, and ddPCR was found to be a more versatile tool over existing qPCR assays, since it yielded more accurate results and was less affected by amplification inhibitors [53].

**Table 1 plants-10-01446-t001:** Molecular assays developed for the general detection of *P. brassicae*.

	Year	Technique	Primer Sequences (5′ to 3′)	
Ito et al. [43]	1999	Nested PCR	Outer	PBTZS-2: CCGAATTCGCGTCAGCGTGA ^a^
			Inner	PBTZS-3: CCACGTCGATCACGTTGCAAT PBTZS-4: GCTGGCGTTGATGTACTGGAA
Faggian et al. [44]	1999	Nested PCR	Outer	PbITS1: ACTTGCATCGATTACGTCCC PbITS2: GGCATTCTCGAGGGTATCAA
			Inner	PbITS6: CAACGAGTCAGCTTGAATGC PbITS7: TGTTTCGGCTAGGATGGTTC
Wallenhammar & Arwidsson [45]	2001	Nested PCR	Outer	PBAW-10: CCCCGGGGATCACGATAAATAACA PBAW-11: GGAAGGCCGCCCAGGACTACC PBAW-12: GCCGGCCAGCATCTCCAT PBAW-13: CCCCAGGGTTCACAGCGTTCAA
			Inner	PBTZS-3 [43] PBTZS-4 [43]
Cao et al. [46]	2007	PCR	TC1F: GTGGTCGAACTTCATTAAATTTGGGCTCTT TC1R: TTCACCTACGGAACGTATATGTGCATGTGA	
Sundelin et al. [47]	2010	qPCR	Pb4-1: TACCATACCCAGGGCGATT PbITS6 [44]	
Rennie et al. [48]	2011	qPCR	DC1F: CCTAGCGCTGCATCCCATAT DC1R: CGGCTAGGATGGTTCGAAAA	
Wallenhammar et al. [49]	2012	TaqMan qPCR	PbF: AAACAACGAGTCAGCTTGAATGC PbR: TTCGCGCACAAGCAC TTG (Probe) PbP: CGCGCCATGCGACACTGTTAAATT	
Cao et al. [51]	2014	TaqMan qPCR	TC1F: GTGGTCGAACTTCATTAAATTTGGGCTCTT RTPbR1a: TCAGCACCGTTTCCGGCTGCTAAGGC (Probe) TCPb1: AAGAAGGAGAAGTCGTAACAAGGTTTC	
Deora et al. [50]	2015	TaqMan qPCR	PBGFPuv3F: CCTAGCGCTGCATCCCATATCGATGGCCCTGTCCTTTTAC PBGFPuv3R: CGGCTAGGATGG TTCGAAAGTGTAATCCCAGCAGCAGTTA (Probe) GFP1: ACCATTACCTGTCGACACAATCTGCCCT	
Al-Daoud et al. [52]	2016	PMA-PCR	PBGFPuv3F [50] PBGFPuv3R [50] (Probe) GFP1 [50]	
Wen et al. [53]	2020	ddPCR	DC1F: CCT AGC GCT GCA TCC CAT AT DC1mR: CGGCTAGGATGGTTCGAAA (Probe) PB1: /56-FAM/CCA TGTGAA/ZEN/CCG GTGACGTGCG/3IABkFQ/	

^a^ Primer PBTZS-2 is used as the sole outer primer for this nested PCR for amplifying the fragment from DNA samples [43].

While essential for clubroot diagnostics, all of these molecular assays were developed for *P. brassicae* detection and cannot differentiate pathotypes. They consist of primers predominantly associated with the conserved regions of the genome, which are fairly similar among pathotypes.

## 3. Rapid Molecular Pathotyping

Molecular diagnostic assays may be developed based on genetic variation between *P. brassicae* pathotypes. Several advantages are presented by molecular-based techniques for pathotyping. They can be highly sensitive and rapid, cost-effective in terms of labor, space, and time, biosecurity is not a concern, and the limitation of inter-rater reliability is eliminated. Various potential molecular markers have been evaluated for pathotyping of *P. brassicae*. A presumptive random amplified polymorphic DNA marker specific to pathotype P_1_, as defined by the differentials of Somé et al. [31], was identified and converted into a sequence characterized amplified region [54]. The *Cr811* gene was found to be specific to pathotype 5 [55], as defined on the differentials of Williams [29]. Over 1500 single nucleotide polymorphisms (SNPs) differentiating two distinct populations of *P. brassicae* were identified and characterized based on the pathogenicity of the isolates and their ability to cause disease against CR canola cultivars [56]. A region of an 18S internal transcribed spacer sequence was found to be specific to pathotype 5X [57], as defined on the CCD set [20], and this region was developed into a probe-based qPCR assay to identify pathotype 5X.

### 3.1. Amplicon Length Distinction

Amplicon length distinction can be used to differentiate pathotypes in PCR-based assays. In this methodology, the insertion/deletion (indel) polymorphism is positioned within the amplicon, and the assay relies on electrophoretic separation of amplified DNA fragments. Pathotype clustering may be performed based on the molecular weight of the band in the gel (Figure 2). This technique was used to distinguish defoliating from non-defoliating pathotypes of *Verticillium dahliae* Kleb., the pathogen responsible for the Verticillium wilt disease of cotton (*Gossypium hirsutum* L.) [58]. The assay was found to be effective in identifying genetic relationships among pathotypes. This rapid technique is advantageous in its use of a single primer pair that is conserved among the tested races/pathotypes, and it is a simple conventional PCR method that can be easily adopted by diagnostic laboratories at minimal cost. However, this technique is dependent on the existence of a discriminatory insertion or deletion and an appropriate conserved region for primer design around the amplicon, and these may not be as abundant as necessary to provide sufficient resolution.

### 3.2. SNP-Based Distinction

Plant pathologists studying a variety of pathosystems have explored the potential of discriminatory polymorphic regions and single nucleotide polymorphisms (SNPs) for pathotype detection. Distinctive primers contain specific polymorphisms corresponding to a subset of pathotypes, resulting in differential PCR amplification (Figure 3). The development of SNP-based assays follows a general three-part process: (1) SNP discovery through sequencing and polymorphism detection by bioinformatics approaches and evaluation on a small sample set; (2) validation of SNPs to eliminate sequencing errors; and (3) assay adoption for pathotyping in large populations [59]. Several molecular techniques have been tested in plant pathosystems and may be suitable for SNP-based pathotyping in clubroot diagnostics.

Distinguishing polymorphic primers were used to explore the pathotype diversity of *Ascochta rabiei* (Kovatsch.) Arx, the fungus responsible for Ascochyta blight of chickpea (*Cicer arietinum* L.), and this assay was able to differentiate three pathotype clusters [60]. Polymorphic primers were also used to distinguish between defoliating and non-defoliating pathotypes of *V. dahliae*, the causal agent of Verticillium wilt of olive (*Olea europaea* L.) [61]. The expected products were amplified when the primers were used independently, although the electrophoretic bands were faint. To increase the sensitivity, the primers were adapted into a nested PCR assay. The nested PCR design increased resolution between defoliating and non-defoliating pathotypes of *V. dahliae*.

In clubroot, six population-specific primer pairs were designed to detect avirulent pathotypes and five were designed to detect virulent pathotypes [62] based on discriminating polymorphic regions identified earlier [56]. The avirulent population refers to pathotypes discovered before the 2009 commercialization of CR cultivars that are known to have been present in Alberta for the longest period. The virulent population refers to pathotypes that emerged due to selection pressure imposed by CR cultivars and which can overcome resistance. One primer pair for each population was further developed into a quantitative assay. All eleven primer pairs were confirmed to be specific to the population they were designed for, and no amplification occurred in non-infested samples. The researchers noted that although the primer pairs are reliable for clubroot diagnostics, they are not as sensitive as earlier reported assays for general detection of *P. brassicae* [62].

Differentiating polymorphic primers provide a rapid and simple conventional PCR-based method that can be easily adopted by a wide range of diagnostic laboratories at minimal cost. Primer pairs for each respective polymorphic cluster need to be designed. Since this is a PCR-based approach, the primers may be optimized into a quantitative assay. There may be a slight increase in running time in carrying out this assay, considering this method involves different primer pairs. The method may be vulnerable to misidentification due to non-specific amplification, false negative results, or technical errors of the PCR. In addition, the longevity of differentiating primers is dependent on the stability of the discriminatory SNPs and the risk for further mutations. SNP-based distinction may also be valuable for a ddPCR assay, a system for absolute quantification used to detect low DNA concentrations [63]. With *P. brassicae*, ddPCR could be used to increase detection of low abundance spores.

### 3.3. RNase-H Dependent PCR (rhPCR)

In conventional PCR, differentiation of pathotypes with slight nucleotide variations is challenging as primers may still bind non-specifically depending on PCR conditions, which may lead to false-positive amplification. A novel primer technology known as RNase-H dependent PCR (rhPCR), introduced in 2011, provides greater accuracy and sensitivity [64]. rhPCR primers are blocked by a single ribonucleotide residue at the discriminating polymorphic site, preventing amplification by the polymerase (Figure 4). The blocked primers are activated via cleavage of the RNA base by the thermostable RNase H2 enzyme once the ribonucleotide residue has annealed to the template strand. In the case of a mismatch, no cleavage will occur, and the primer remains blocked.

The rhPCR technique has been incorporated into the diagnostic process of several pathosystems [65,66,67] and for whitefish species identification [68]. An rhPCR assay was developed to identify native from invasive subspecies of common reed (*Phragmites australis* (Cav.) Trin. ex Steud.) [65], showing correct identification of the subspecies with their respective primer pair, and not with the opposing primer set. No amplification occurred in an rhPCR test without the use of the RNase H2 enzyme, confirming the effectiveness of the block-cleavable technology. In the same study, rhPCR was compared with a previously used restriction fragment length polymorphism (RFLP) protocol. While the level of accuracy was similar, rhPCR was less time-consuming and easier to perform [65]. Another rhPCR assay was developed for the identification and quantification of *Grosmannia clavigera* (Robinson-Jeffrey & R.W.Davidson) Zipfel, Z.W. de Beer & M.J. Wingf., a fungal pathogen of pine trees vectored by mountain pine beetles (*Dendroctonus ponderosae* Hopkins) [66]. The quantitative rhPCR assay was able to efficiently and accurately distinguish *G. clavigera* from other species, highlighting the potential of this technique to be used in the diagnostic process of complex phytopathogenic samples. The sensitivity of the rhPCR technique was discussed during the development of a pathotyping assay to discriminate isolates of *Salmonella enterica* subsp. *serovar* Heidelberg (SH), a bacterium responsible for salmonella infection in humans [67]. The assay improved resolution and efficiency in isolate discrimination when evaluated against pulsed field gel electrophoresis (PFGE) and phage typing. While the rhPCR pathotyping results were generated as early as 5 h after DNA extraction and isolates were accurately identified, non-specific amplification of isolates containing a non-targeted alternate allele at the discriminatory SNP position occurred in 15% of the reactions. However, these non-specific amplicons were distinguishable from amplicons of targeted isolates via a difference in band intensity. An RNase H2 enzymatic activity error consistent with this result has been previously noted [64]; inaccurate RNase H2 cleavage may occur in a mismatch in the target sequence, but at a much lower frequency relative to the target match. Although not part of a pathosystem diagnostic process, a quantitative rhPCR assay was used for the detection of five closely related whitefish (*Coregonus*) species [68]. Detection had been carried out previously with conventional TaqMan qPCR for other *Coregonus* species; however, sequence variation between these five closely related species was not strong enough for a conventional qPCR assay design. The two techniques were combined to increase specificity by incorporating rhPCR primers into the TaqMan assay. To ensure amplification was specific and to evaluate the rhPCR technique, the TaqMan assay was run in parallel with conventional primers. The addition of the blocked-cleavable technology increased specificity, as no non-targeted amplification occurred.

In clubroot, an rhPCR assay was developed to differentiate a new virulent ‘pathotype 3-like’ strain of *P. brassicae* from the original pathotype 3 [69], as defined on the differentials of Williams [29]. Based on polymorphic regions of two hypothetical protein genes, two primer pairs were designed as a duplex PCR with one primer pair for each gene, and each gene representative of either pathotype 3-like or 3. The primers corresponding to pathotype 3-like produced an amplicon of 135 base pairs, whereas the primers for pathotype 3 produced a larger amplicon of 337 base pairs. The sensitivity of the primers was tested against pre-pathotyped single-spore and field isolates of the pathogen. Equal proportions of pathotype 3-like and pathotype 3 DNA produced bands of comparable intensity. When the pathotype 3-like DNA proportion was greater, its respective band was stronger relative to pathotype 3. Likewise, a greater proportion of pathotype 3 DNA produced stronger bands relative to pathotype 3-like. The researchers tested the assay against four pre-pathotyped field galls representing different counties in Alberta. Each gall produced amplicons with both rhPCR primer pairs, suggesting that field galls are a mixture of virulent and avirulent pathotypes. With the same rhPCR assay, the assay was tested on 79 field galls collected from 22 fields in Alberta [70]. *P. brassicae* populations from 50 of these galls produced more than one band, confirming the hypothesis that multiple pathotypes co-exist as field populations in a single field gall and that their abundance varies according to their interactions with host plants. The galls were not subjected to phenotypic pathotyping, and therefore the exact pathotypes responsible for the bands were not confirmed.

The above studies demonstrate the potential of the rhPCR technique in SNP discrimination of *P. brassicae* pathotypes. rhPCR offers greater resolution and sensitivity in comparison with previously established assays. The combination of rhPCR with other molecular techniques may further increase its SNP-differentiating ability. For instance, rhPCR primers may be designed against a region bearing a higher number of differentiating SNPs to combine the technology with the previously discussed SNP-based distinction. This may increase the reliability and sensitivity of the pathotyping assay. As with most PCR-based assays, the rhPCR primers may be optimized into a quantitative assay. Other than the additional RNase H2 enzyme required, the protocol can be easily adopted by diagnostic laboratories, considering rhPCR is carried out with the same equipment as conventional PCR. The turnaround time is slightly longer in comparison with assays using only one primer pair.

### 3.4. Single Base Extension (SBE)

Another SNP-based technology is SNaPshot [71], a single base extension (SBE) reaction that allows the detection of multiple SNPs on multiple DNA templates [59]. SNaPshot primers are designed just upstream of the polymorphic base in question. When the primers anneal to the DNA template right before the SNP, Taq polymerase extends the primer by one nucleotide on the 3′ end by selecting the correct complementary base from a pool of fluorescently labelled dideoxy nucleotide triphosphates (ddNTPs), which impairs further extension of the product. The incorporation of the pathotype-specific base at the 3′ end produces a fluorescent signal corresponding to one of four dyes that match each of the four possible bases. The resulting product size is the length of the SNaPshot primer plus the addition of the fluorescent ddNTP base. The SBE product is then separated by capillary electrophoresis inside a genetic analyzer to generate electropherograms, and pathotypes are identified based on peak color and product size.

The SNaPshot protocol is a four-step approach: (1) template preparation; (2) extension reaction; (3) post-extension treatment; and (4) capillary electrophoresis (Figure 5). In template preparation, targeted conventional PCR amplification is used to generate the DNA templates containing the SNP. The resulting templates must undergo a purification process to remove PCR primers and unincorporated deoxynucleoside triphosphates (dNTPs) to avoid interference with the extension reaction. The extension reaction takes place, followed by a post-extension treatment of the products. The products are incubated with either shrimp alkaline phosphatase (SAP) or calf intestinal phosphatase (CAP) to remove any unincorporated ddNTPs. To prepare for capillary electrophoresis, purified SNaPshot products and size standards are added into a Hi-Di formamide solvent. DNA size standards are used to determine the size of the SNaPshot product. Capillary electrophoresis is conducted inside a genetic analyzer, and the results of this scan are examined using a fragment analysis software.

The SBE technique has been incorporated into the diagnostic process of several different pathosystems. A SNaPshot assay was developed to detect variants of the Potato virus Y (PVY), a pathogen belonging to the *Potyvirus* genus and a threat to crops of the Solanaceae family [72]. Similar to *P. brassicae* pathotyping methods, PVY variant identification is usually carried out phenotypically based on disease symptoms on host plants. The SNaPshot technique was advantageous in its ability to characterize variants of mixed samples, and in the amount of starting material required. Reliable detection occurred with as few as 10^2^ copies of the PVY genome [72] in comparison with 10^3^ copies required in a previously published real time PCR assay [73]. An assay was also designed for the specific detection of race 3 of *Fusarium oxysporum* f.sp. *vasinfectum* (FOV), a soilborne fungal pathogen responsible for root rot, vascular wilt, damping-off, and yellowing in a wide range of economically important crops [74]. While the SNaPshot technology has not been evaluated for *P. brassicae* pathotyping, it has been used to detect erucic acid, a fatty acid in canola, for marker-assisted selection in canola breeding [75]. Canola plants with a two-base deletion in one specific gene produced nominal erucic acid content. This polymorphic locus was used for the development of a SNaPshot assay. It was necessary to develop an assay to identify erucic acid content rapidly at an early plant growth stage, since high levels of erucic acid reduce oil quality and digestibility.

The above studies highlight the reliable and scalable potential of incorporating SNaPshot as a *P. brassicae* pathotyping tool. Several advantages of the SBE technique were consistently noted for the developed assays. The procedure is straightforward and the automation of genetic analyzers offers convenient data processing with a high degree of accuracy. In comparison with the previously mentioned SNP-based PCRs, where amplification proceeds based on the existence of a specific allele, SNaPshot allows for the detection of up to four allelic variants as bases are distinguished by means of fluorescent ddNTPs. SNaPshot is scalable through a multiplex reaction in which numerous polymorphic regions in the genome can be examined concurrently for efficient and rapid testing. Nonetheless, there are several limitations to this technique. It is a lengthy process due to the number of steps involved, and it requires equipment that may not generally be used by diagnostic laboratories. In addition to basic PCR reagents, a clean-up kit is needed to purify PCR products, SNaPshot reagents are necessary for the extension reaction, shrimp alkaline phosphatase or calf intestinal phosphatase is needed for post-extension treatment, Hi-Di Formamide is required as an injection solvent in the genetic analyzer, and a size standard is required to investigate the results of the fluorescent peaks.

### 3.5. Can Metabarcoding Be Used in Clubroot?

The advent of next generation sequencing (NGS) and bioinformatics platforms has enabled new lines of research by groups studying a variety of plant pathosystems. NGS services were introduced as a highly efficient and sensitive sequencing platform that overcame the limitations of Sanger sequencing with respect to throughput, making it an appealing strategy for pathogen analyses [76]. NGS has greater sensitivity to detect low-frequency variants, as it has been able to detect pathogens that were not detected by other molecular tests, especially in early stage infections [77]. By isolating and sequencing small RNAs from co-infected plants using Illumina deep sequencing technology, both the sweet potato feathery mottle virus and the sweet potato chlorotic stunt virus were detected in sweet potato (*Ipomoea batatas* (L.) Lam.) [78], and multiple viruses infecting a single ornamental plant were detected [79]. Double-stranded RNA from leaves of infected apple plants (*Malus domestica* Borkh) were isolated and sequenced using Illumina technology, resulting in the identification of 12 genotypes of the apple stem pitting virus [80]. Full genome sequencing was found to be superior in detecting both early stages and low levels of infection by viral pathogens in grapevine (*Vitis vinifera* L.), in comparison with bioassays that are dependent on disease symptoms [77]. Collectively, these studies indicate that NGS technologies allow for the simultaneous detection of multiple pathogens [78,79] or pathotypes [80] in a single sample, and the detection of low-frequency genomes [77]. Full genome sequencing allows for the generation of substantial amounts of data to discover variability among pathotypes, which can be used for diagnostics and metagenomics studies.

Metabarcoding is an NGS approach that offers high quality single nucleotide resolution in a single reaction [81], and may be adopted into the diagnostic system for *P. brassicae*. The alignment of full pathotype genomes may reveal candidate loci that could be used as pathotype distinctive barcodes for the development of metabarcoding assays. This type of assay may generate masses of short reads of identifying barcodes for *P. brassicae* pathotype characterization, simultaneously detecting multiple pathotypes from a clubroot sample via amplicon-based targeted sequencing. The resulting number of short reads may be proportional to the pathotype composition of a sample. While this may not be absolute quantitative data, it may offer insights on relative pathotype abundance [82]. The technique can also provide a representation of the pathotype diversity present in a sample. The identifying barcodes would be designed based on discriminative polymorphic regions. To be functional as a barcode, the region must have sufficient pathotype-level genetic variability, conserved flanking sites for universal primers, and a short enough sequence length for amplification [83]. It is essential to subject the primers to a comprehensive BLAST search to ensure specificity to the barcoding region and to *P. brassicae*, and to validate amplification through Sanger sequencing with a select number of samples before mass sequencing.

The protocol is a six-step process: (1) PCR; (2) preparation of sequencing libraries [84]; (3) NGS; (4) filter sequencing reads; (5) sequence assembly; and (6) pathotype identification (Figure 6). An initial round of conventional PCR generates the barcoded amplicons and increases the number of DNA copies to be sequenced. Each sample can be labeled with a DNA tag during PCR to identify the origin of sequencing reads [85]. Multiplexing through DNA tagging allows hundreds of samples to be processed in one sequencing run. Tagged amplicons are then constructed into a sequencing library in preparation for NGS; indexing adapters of known sequences are annealed to the amplicons and the library undergoes a final quantification. Once sequenced, low quality reads are filtered out to reduce sequencing errors and increase accuracy [86]. The resulting sequencing reads are then assembled and aligned to reference barcodes that are typical of each pathotype, revealing the pathotype of the samples. Incorporating probe-based capture may increase the efficiency of NGS in detecting specific pathotypes. This occurs through the hybridization of a probe designed specifically for a targeted sequence representing a particular pathotype, with the resulting hybridization signal indicating the recognition of the targeted pathotype [87,88,89].

Metabarcoding has been proposed and evaluated as a platform for diagnostics in plant pathosystems [90]. It was used to analyze the fungal spore composition in air samples and found to detect a much wider range of pathogens relative to previous methods [91,92]. Earlier approaches were insufficient as they could only detect a small fraction of the total fungal diversity. Metabarcoding was also used to investigate *Fusarium* species composition, and it was able to detect 17 species in soil samples and maize (*Zea mays* L.) residue [93]. The ability to pool hundreds of samples in one sequencing reaction, efficient turnaround time, and increased accuracy were highlighted. Another study tested this approach with *Phytophthora* species in artificially infested soil samples, and found that the sequencing results were comparable with the soil composition [94]. Metabarcoding was shown to provide realistic approximations of species abundance when it was used to characterize *Colletotrichum* species on walnut (*Juglans regia* L.) [95]. While metabarcoding has not been established for the clubroot pathogen, an exploration of this method is underway in our research group. We are refining genomic assemblies for 45 single-spore and field isolates, which will allow us to look for rapidly evolving discriminative polymorphic regions to select for metabarcoding for use in *P. brassicae* pathotyping.

Several limitations are associated with metabarcoding as a diagnostic tool. Bioinformatics expertise and access to bioinformatics facilities are required to analyze the sequencing data competently. Contamination of samples is of greater concern due to the high sensitivity of NGS platforms. Therefore, thorough BLAST testing of the primers against a wide set of microorganisms and preliminary conventional PCR testing of the primers against other species are essential to ensure specificity for the target. Pathotypes in the sample may not be uniformly amplified, as the generation of barcoded amplicons is dependent on conventional PCR. This was evaluated with parasitic soil protist communities and results provided estimations of relative abundance different from those expected [96]. Misrepresentation may also occur during sequencing; three different NGS platforms produced different sequencing outputs when quantitation of artificially assembled fungal communities was evaluated [97]. Due to this, it can only provide approximate pathotype proportions. Nonetheless, with the widespread adoption of genome sequencing in plant pathology research and the identification of variant information, and with the flexibility of these methods for a wide range of experimental designs, metabarcoding and NGS technologies may play a substantial role in clubroot diagnostics.

### 3.6. General Limitations of Molecular Approaches

While molecular techniques are the future of clubroot diagnostics, there are some limitations to take into consideration. A possible challenge is finding polymorphisms that provide consistent genomic and phenotypic clustering of pathotypes. The extensively used CCD set and other host differential systems are based on phenotypically distinctive virulence patterns, which may not always be in agreement with DNA sequence variations. In the case of disagreements, pathotype classifications may be modified to incorporate the genomic data or differentiating polymorphic regions must be deliberately chosen to distinguish pathotypes as defined by the CCD or other differential sets. The development of molecular diagnostic assays is dependent on comprehensive sequence databases for the discovery of suitable polymorphic regions, and it would be beneficial to have a reliable genome for each individual pathotype. It may be challenging to assemble variable regions by a reference-based assembly approach, and therefore generating full de novo assemblies with long-read technology mixed with short reads may be necessary. Polymorphic regions with the greatest level of diversity will offer the greatest differentiating capacity. Insufficient genetic variability among pathotypes may cause complications in assay design and therefore restrict the use of molecular-based techniques. In addition, defining a unique sequence for each individual pathotype may not be possible in the same region of the genome. Multiple regions may need to be used to thoroughly differentiate the pathotypes. Primers must be specific enough to avoid amplification of non-target microorganisms and non-target regions of the genome. In the case of amplicon length distinction and metabarcoding, it is imperative for the conserved primers to be generic enough to amplify the DNA of all pathotypes. The cost of adopting new molecular diagnostic tools is also an issue. However, due to the increasing interest in molecular approaches, prices may follow a downward trend as these techniques become more widely used and taken up by diagnostic laboratories.

## 4. Future Perspectives

Several PCR-based, SNP-based, and sequencing technologies have been introduced into diagnostic processes of plant pathosystems (Table 2). PCR remains the most cost-effective and most widely used molecular technique. Several PCR-based methods may be modified into a quantitative assay, which can allow evaluation of inoculum levels or degree of host colonization. rhPCR has been shown to further increase sensitivity of SNP discrimination over conventional PCR. However, the developmental phase of PCR-based assays may require a lengthy standardization process since numerous factors must be considered to minimize non-specific amplification. The main advantages of SNaPshot are its ability to multiplex samples and to distinguish any of the four alleles at the discriminatory SNP; however, this technique requires higher operational costs and is not quantitative. NGS-based metabarcoding has the greatest sensitivity and scalability; however, it is not absolutely quantitative and routine NGS will require in-depth post-sequencing bioinformatics analysis.

An ideal clubroot diagnostic tool is a mixed strategy of techniques to further increase the sensitivity and accuracy of the assay. An integrated process of metabarcoding, SNP-based distinction, and rhPCR may give the most comprehensive depiction of clubroot samples. Metabarcoding would be used initially for an assessment of pathotype diversity in a clubroot sample because of its conserved primers and high resolution. Once pathotypes are identified, a qPCR assay combining SNP-based distinction and rhPCR would indicate pathotype abundance. In this case, only primers specific to the pathotype(s) identified by metabarcoding are required.

## 5. Conclusions

The common goals of increasing production and sustaining economic security are recognized within all sectors of the canola industry. As with every disease, an important criterion that contributes to its economic, ecological, and social consequence is the virulence of the causal agent. Due to the emergence of new virulent pathotypes of *P. brassicae* and decreased effectiveness of resistant cultivars, clubroot management has increased in complexity. Each described molecular pathotyping approach has its own advantages and limitations, as they differ in sensitivity, scalability, accessibility and operational costs. Efforts to standardize a comprehensive diagnostic system will progress in parallel with the generation of improved *P. brassicae* reference databases. Clubroot researchers in search of molecular markers for pathotype detection will make their share of contributions towards the development of future *P. brassicae* diagnostics, as rapid molecular pathotyping assays are dependent on sequence variations and polymorphic regions. Overall, the technique of choice for a rapid molecular pathotyping tool should be accessible by clubroot diagnostic laboratories.

## Figures and Tables

**Figure 1 plants-10-01446-f001:**
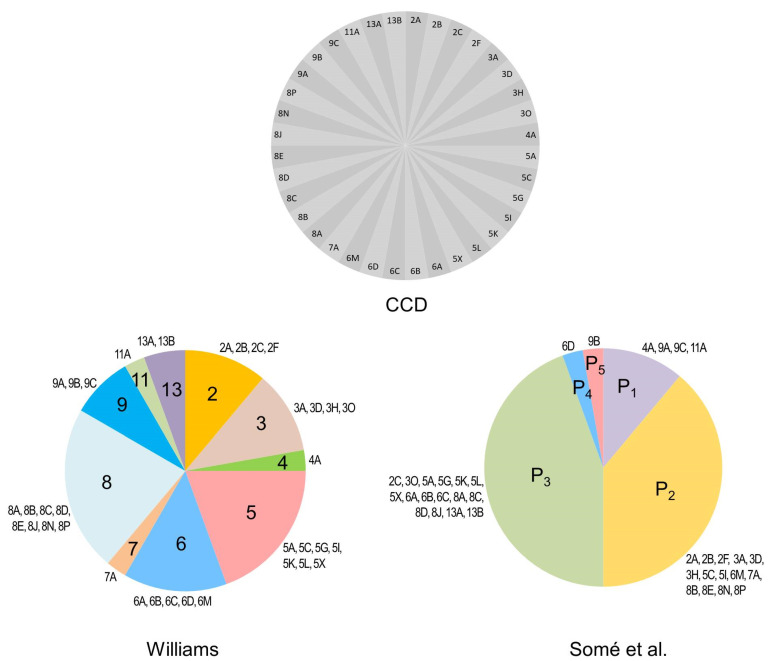
Designation of *P. brassicae* pathotypes from Canada as defined on the Canadian Clubroot Differential (CCD) set [20] in comparison with their classification on the systems of Williams [29] and Somé et al. [31]. Slices of each pie chart for Williams and Somé et al. denote the proportion of each respective pathotype designation by its representation in Canada, with the corresponding CCD pathotypes indicated. Since the CCD set includes the differentials of Williams and Somé et al., it is possible to obtain all respective pathotype designations based on the reaction of the CCD hosts. Pathotypes 2B, 2F, 3A, 3D, 3H, 3O, 5C, 5G, 5I, 5K, 5L, 5X, 6M, 8E, 8J, 8N, and 8P were first reported by Strelkov et al. [20]. Pathotypes 2A, 4A, 6A, 6B, 6C, and 7A were first reported by Askarian et al. [23]. Pathotypes 5A, 8A, 8B, and 8C were first reported by Strelkov et al. [22]. Pathotypes 2C, 6D, 8D, 9A, 9B, 9C, 11A, 13A, and 13B were first reported by Hollman et al. [21].

**Figure 2 plants-10-01446-f002:**
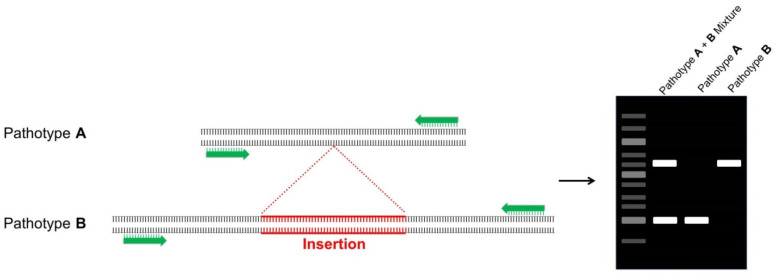
This figure summarizes the amplicon length distinction method. One primer pair (shown in green) is used for the detection of both pathotypes, as it is designed against a conserved region. The hypothetical pathotype B has a distinctive insertion (shown in red) within the sequence that will produce a greater amplicon size in comparison with hypothetical pathotype A. The electrophoretic gel presents a noticeable difference in molecular weight of the pathotypes. Both pathotypes are detectable as the polymorphism is located within the amplicon.

**Figure 3 plants-10-01446-f003:**
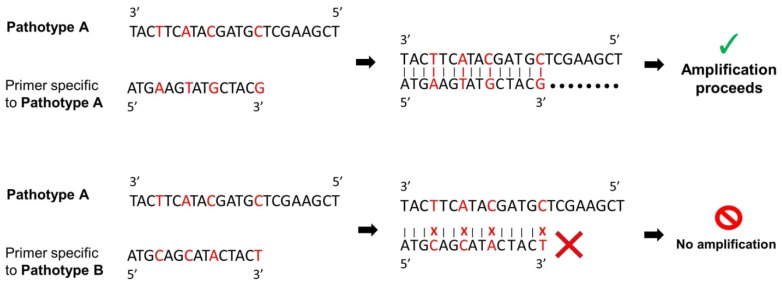
This figure summarizes the SNP-based distinction method. Primers are designed to target a distinctive polymorphic sequence specific to a pathotype or pathotype cluster. When a primer specific to the hypothetical pathotype A is used against hypothetical pathotype A, amplification occurs because of the perfect match of primer and DNA template. However, when a primer specific to hypothetical pathotype B is used against hypothetical pathotype A, mismatches prevent amplification.

**Figure 4 plants-10-01446-f004:**
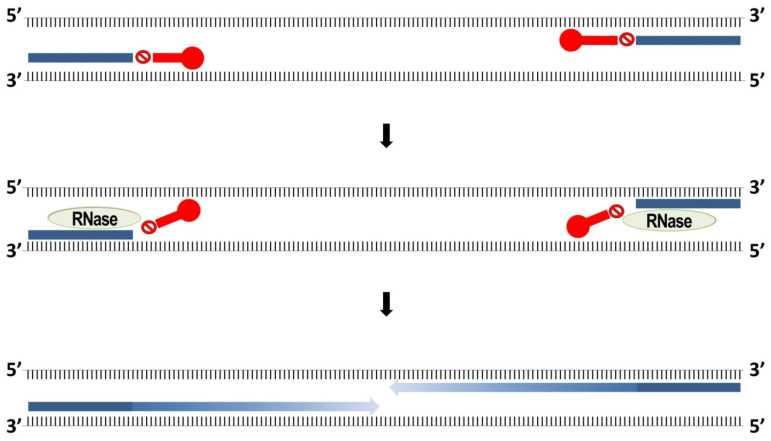
This figure summarizes RNase-H dependent (rhPCR). The rhPCR primers are blocked by a ribonucleotide residue as represented by the stop symbol, followed downstream by 4 DNA bases complementary to the template as shown in red. The red circle at the 5′ end of the blocked primer is a propanediol C3 spacer. Once the RNase H2 enzyme comes in and cleaves off the ribonucleotide residue, the primers are activated and extension by DNA polymerase continues.

**Figure 5 plants-10-01446-f005:**
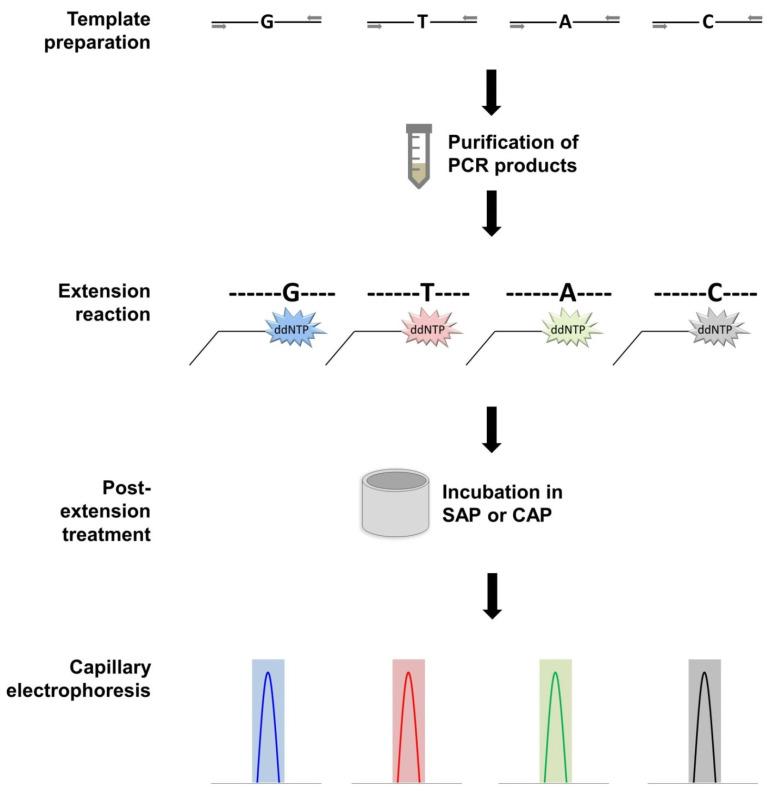
This figure summarizes the workflow of the single base extension (SBE) technique. Samples undergo an initial PCR to generate templates encompassing the single nucleotide polymorphism (SNP). Once templates are purified, the extension reaction occurs. Allele-specific fluorescently labelled ddNTPs are matched to the SNP. Extension products are incubated in either shrimp alkaline phosphatase or calf intestinal phosphatase to remove unincorporated ddNTPs. Products are scanned via capillary electrophoresis in a genetic analyzer and differentiating SNPs are revealed in the resulting electropherograms. The G fluoresces blue, T fluoresces red, A fluoresces green, and C fluoresces black.

**Figure 6 plants-10-01446-f006:**
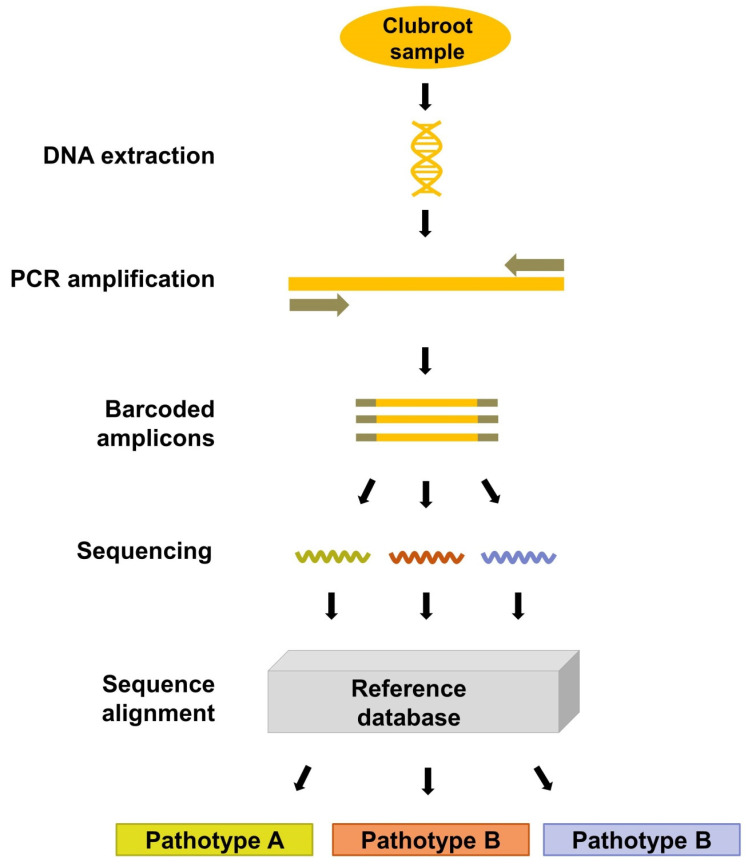
This figure summarizes the metabarcoding workflow. Genomic DNA is extracted from *P. brassicae* samples and undergoes an initial PCR to generate barcoded amplicons. Amplicons are prepared for next generation sequencing. The resulting sequencing reads are aligned to the reference barcode database to identify the pathotypes in the sample.

**Table 2 plants-10-01446-t002:** Comparison of proposed pathotyping platforms for *P. brassicae*.

Technique	Efficiency ^a^	Specificity	Quantitative Potential	Primers Required	Costs	Main Advantages	Main Disadvantages
Amplicon length distinction	Low	Low	No ^b^	1 pair per indel	Low	Conserved primers	Low scalability
SNP-based distinction	Low	Low	Yes	1 or 2 pairs per polymorphic region ^c^	Low	Simple procedure	Low scalability and sensitivity
rhPCR	Low	Moderate	Yes	1 or 2 pairs per polymorphic region ^c^	Moderate ^d^	Simple procedure	Low scalability
SBE	Moderate	Moderate	No	1 pair + 1 SBE primer per SNP ^e^	High	Scalable; can detect any allele	Non-quantitative; lengthy procedure
Metabarcoding	High	High	Partially ^f^	1 pair per barcoding sequence	Very High	High sensitivity and scalability	High costs and expertise required; lengthy procedure

^a^ The scalability and throughput ability of the technique. ^b^ Primers are designed against a conserved region with the indel positioned within the amplicon (Figure 2). A qPCR assay would not identify the pathotype since amplification occurs regardless of pathotype under investigation. ^c^ While only one primer pair is required to identify the SNPs, a second primer pair of alternate alleles would further verify pathotype detection. ^d^ RNase H2 enzyme and its dilution buffer is required in addition to basic PCR reagents [98]. The price for rhPCR primers is slightly higher than conventional primers [99]. ^e^ A conserved primer pair is required to generate the template for the SBE reaction in addition to the SBE primer (Figure 5). ^f^ Metabarcoding cannot provide absolute quantities, only estimates of pathotype proportions.

## Data Availability

Not applicable.

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
