# Peer review of "Current and Future Pathotyping Platforms for Plasmodiophora brassicae in Canada"

_plants, 2021, doi:10.3390/plants10071446_

Round 1

Reviewer 1 Report

The review of Heather H. et al., Current and Future Pathotyping Platforms for Plasmodiophora brassicae is devoted to the possibility of using modern methods of molecular biology in the pathotyping of Plasmodiophora brassicae - one of the malicious pathogens of the cabbage family that causes cabbage keel. The disease caused by this pathogen is classified as a quarantine object and the need for pathotyping is due to the fact that the definition of the disease by traditional visual methods occurs at almost the last stages of the development of the disease and the results of such work can only speak of a fait accompli. Methods of molecular diagnostics allow all diagnostic and pathotyping procedures to be carried out at an earlier date, which on the one hand accelerates the mapping of the spread of pathogens, and on the other, gives time to choose the most effective way to combat this pathogen.

1)  Paragraphs 1.1-2.1 give information on P. brassicae spreading, management and pathotypes in Canada. Authors should provide data on P. brassicae situation in the whole world, or change the title on “Current and Future Pathotyping Platforms for Plasmodiophora brassicae in Canada”.

2)  In the Abstract authors wrote that “Early, preventative pathotype detection is imperative to maximize productivity and promote sustainable crop production”. This thesis should be discussed further below in the section on this pathogen management (for example).

3)  I strongly encourage adding a table including sequences of PCR primers, which was used for P. brassicae detection (section 2.3. Molecular Approaches).

4)  Figures 2- 6 illustrate schemes of different molecular methods, but these methods aren’t different for P. brassicae and other living organisms. The article cannot include the information that may occur in tutorials. These figures should be removed. I propose to summarize differentiations of methods which refer only to P. brassicae (or plant pathogens) in figure or table (like advantages and disadvantages of methods summarized in table 1).

5)  If molecular methods were used in Canada, the information on comparison of the results of these investigations with morphological pathotyping is of great interest. Could authors provide it in this article?

6)  In the section “Metabarcoding” it seems like this method can be used only for clubroot pathogen (no other organisms) identifying (“Metabarcoding is a NGS approach that can simultaneously identify multiple pathotypes from a clubroot sample via amplicon-based targeted sequencing”). But this section does not include some results on P. brassicae. It should be fixed.

7)  A part of the section “Future perspectives” seems like a shopping list: “Bioinformatics software and training is required for metabarcoding. If qPCR is not already utilized in a laboratory, qPCR instruments, associated reagents and software for downstream analysis also need to be obtained. As well, there are additional costs associated with purchasing qPCR primers and the RNase H2 enzyme”.

Authors should remove it and discuss only methodology perspectives.

8)  Reference sources must be found and added in the manuscript.

I recommend publishing this article after improving.

Reviewer 2 Report

This is a timely comprehensive review for detecting Plasmodiophora in clubroot disease of canola, which is largely well written.

However, mentioned below are some corrections, that need to be incorporated before the manuscript can be considered for publication.

  1. Page 3 - section 2.1. Phenotypic approaches – “pathotype characterization against earlier sets is possible with the CCD set Er-ror! Reference source not found.Error! Reference source not found.(Error! Reference source not found.Error! Reference source not found.Error! Reference source not found.Error! Reference source not found.Error! Reference source not found.).”
  2. Page 7 - section 3.1 – Amplicon length distinction - Pathotype clustering may be performed based on the molecular weight of the band in the gel (Error! Reference source not found.).
  3. Page 7 - section 3.2 – SNP-based detection - Distinctive primers contain specific polymorphisms corresponding to a subset of pathotypes, resulting in differential PCR amplification (Error! Reference source not found.).
  4. Page 9 - rhPCR primers are blocked by a single ribonucleotide residue at the dis-criminating polymorphic site, preventing amplification by the polymerase (

).Error! Reference source not found.

  1. Page 11 - The SNaPshot protocol is a four-step approach: (1) template preparation; (2) exten-sion reaction; (3) post-extension treatment; and (4) capillary electrophoresis (Error! Ref-erence source not found.Error! Reference source not found.).
  2. Page 14 - The protocol is a six-step process: (1) PCR; (2) preparation of sequencing libraries [84]; (3) NGS; (4) filter sequencing reads; (5) sequence assembly; and (6) pathotype iden-tification (Error! Reference source not found.).
  3. Page 16 - 4. Future perspectives - Several PCR-based, SNP-based, and sequencing technologies have been introduced into diagnostic processes of plant pathosystems (Error! Reference source not found.).

Table 1: Legend correction. Also, this table is not mentioned in the text.

Figures: None of the figures are mentioned in the text for reference. Please include them at the appropriate associated text.

Round 2

Reviewer 1 Report

Authors provided a point-by-point response to each of my comments and improved the article. Authors should check figures 2-6: if schemes were taken from tutorials (on another literature), it should be cited. In my opinion, the article can be published without further changes.